



# Transdisciplinary co-production of knowledge for effective flood risk management

Mariele Evers, [1]Britta Höllermann[2], Sylvia Kruse[3]

[1] Department of Geography, University of Bonn, 53115 Bonn, Germany
[2] Institute for Geography, University of Osnabrück, 49074 Osnabrück, Germany
[3] University of Freiburg, Chair of Forest and Environemtal Policy, 79106 Freiburg, Germany

*Correspondence to*: Mariele Evers (mariele.evers@uni-bonn-de)

**Abstract.** Flood risks represent one of the most pressing global challenges, exacerbated by factors such as climate change, ur-
banization, and land use changes. Effective flood risk management (FRM) faces significant challenges, including the need for robust decision-making, addressing existing risks, and implementing strategies across the disaster risk reduction (DRR) cycle. This paper highlights the role of transdisciplinary (TD) research in tackling these challenges, particularly through the co-production of knowledge between scientific and non-scientific actors. Specific characteristics and requirements for flood risk research, which should be considered in TD research, are described. The paper explores three main objectives: (1) illustrating a methodo-
logical design for TD research in flood risk research, (2) applying and expanding the framework of impact generation mechanisms in knowledge co-production, and (3) reflecting on the lessons learned from North-South collaboration in flood risk research. The findings are based on and illustrated with the approach, methods and tools applied and exemplified by a flood risk research project in Ghana, the PARADeS project. The results demonstrate that key mechanisms, such as promoting systems knowledge, fostering social learning, and enhancing leadership competencies, are critical for generating impact. Additionally,
mediators like joint research formulation, trust-building, and anchoring project results were identified as essential for effective implementation and sustainable transformation towards effective DRR. The study concludes that a combination of these mechanisms and mediators, applied contextually, can significantly enhance the effectiveness of flood risk management strategies and contribute to the development of tailored, context-sensitive approaches.

## 1 Introduction

Flood risks are among the most serious of all risks, and severe flooding events are increasing worldwide due to many interlinked factors, among them climate change-induced changes in precipitation patterns, changes in land use and vegetation cover, soil degradation, and an increase in urbanisation processes (Tabari, 2020; IPCC, 2021). At the same time, flood risk management (FRM) faces three significant challenges: first, the need for a sound knowledge base on which to build robust decisions for disaster risk reduction (DRR) along the DRR cycle (i.e., mitigation, emergency response, recovery, and prevention) (Alexander, 2020);
second, the development of strategies and measures as well as robust decisions that address both existing risks and consider inherent uncertainties of knowledge (Mens et al., 2011); and third, the implementation of DRR and flood management strategies and action plans within the different phases of the DRR cycle (Rötzer et al. 2023).

Accordingly, research on FRM and DRR needs to consider these challenges, similar to other fields of sustainability transformation where research is expected to produce a sound knowledge base and support the development of robust decisions and implementa-
tion of strategies and action plans in FRM. Transdisciplinary (TD) research, specifically the co-production of knowledge, is often considered to help to overcome or at least to tackle these challenges. Transdisciplinarity is regarded as a research approach that





involves inter-scientific cooperation between various disciplines and cooperation between science and society by including practitioners and non-scientific actors in the research project (Jahn et al., 2012). In doing so, TD research is expected to address complex societal problems by enabling processes of mutual learning in which knowledge is co-produced between science and society,

integrating various knowledge claims to solve societal or real-world problems (Jahn et al., 2012; Lang et al., 2012; Norström et al., 2020).

While the scientific discourse on TD research was developed mainly by European and North American scholars or has been applied in case studies in those regions in the last 40 years (Brandt et al., 2013), TD research by African scholars and in African cases has increased significantly since 2015, in many cases in North-South projects. Along with this expansion of collaborative and trans-

disciplinary North-South research in the field of sustainability research, it is necessary to elaborate the specific characteristics that go along with TD North-South collaboration (Pärli et al., 2022a; Schneider et al., 2023; Tribaldos et al., 2020). Among the most important characteristics of TD North-South projects are that they tend to focus more on creating societal effects, to include a broad variety of stakeholders, and to result in more prominent knowledge uptake in practice and policy than North TD projects (Pärli et al., 2022a) .

Water has long been identified as a field where various societal needs face complex natural processes that are impacted by multiple drivers of change, thus calling for participatory and TD research (Hedelin et al. 2017, Krueger et al., 2016), and some scholars have already developed TD methods that are specifically useful within water-related research (Renner et al., 2013; Schneider and Rist, 2014). Some TD research on flood risk has even been conducted in North-South collaboration (Almoradie et al., 2020; Culwick and Patel, 2017; Lienert et al., 2022). Almoradie et al. (2020) and Lienert et al. (2022) came to the conclusion that co-design

of the research project is essential and that the main benefit of the respective participatory approach lies in co-production of knowledge.

In a meta study of 31 case studies from TD projects, Schneider et al. (2019) identified key mechanisms in TD research that sought to generate impact. They aimed to identify theories of change through common patterns and lessons learnt. The three generic mechanisms they deduced for impact generation are a) promoting systems, target, and transformation knowledge for more informed

and equitable decision-making, b) fostering social learning for collective action, and c) enhancing competences for reflective leadership.

With our contribution, we intend to expand this field of TD water research; we pursued three objectives: 1.) to reflect on TD research and to illustrate a possible methodological design including possible tools and methods using a research project in Ghana with a focus on specific needs of flood risk (FR) research and North-South collaboration. 2.) to apply and to discuss the framework

of generic mechanisms of impact generation within knowledge co-production processes as developed by Schneider et al. (2019) and to scrutinize mechanisms specific for flood risk-related research in North-South collaboration and further develop the framework by integrating the importance of mediators for TD impact generation, and 3.) to reflect generally on TD research against the background of FR research and on the lessons we can learn from North-South collaboration.

Our case study is part of the research project on FRM in Ghana - the PARADeS project (for more information please see

https://www.geographie.uni-bonn.de/parades/de) which is described in more detail in section 3.1.



## 2 State of research and implications for a transdisciplinary research design of flood risk research

### 2.1 Characteristics of flood risk research

Flood risk research is highly complex. Different flood risk factors on different spatio-temporal scales influence the generation of floods, and exposure and vulnerability affect the actual resulting risk. In the following, we describe the characteristics of flood risk

and the implications these present for TD research.

*Complexity*

Flood risk is a product of interacting physical and societal elements. We define a hazard as the potential occurrence of a certain event. Risk implies that a hazardous event could have negative effects such as damage or fatalities (Cardona et al., 2012). To be effective, risk management needs to be based on a sound understanding of the controlling risk drivers Kreibich et al. (2022). Flood

risk assessments require different disciplines to understand and model the underlying components hazard, exposure, and vulnerability (Sieg et al., 2023; Fuchs et al. 2024). Di Baldassarre et al. (2018) highlight that a focus on just one aspect of flood risk can cause a systematic bias in the selection or prioritization of alternative strategies for flood risk reduction and that we can draw from new approaches that have been developed for the study of socio-nature interactions in various interdisciplinary fields (Di Baldassarre et al., 2018). We see flood risk as a system whose components interact, are dependent in multiple ways, leading to non-

linearity and randomness. Ladyman et al. (2013) describe a complex system with distinct properties that arise from these relationships, such as nonlinearity, emergence, adaptation, and feedback loops. The complexity of flood risk problems calls for broad and cross-cutting interdisciplinary expertise. Thus, a diversity of disciplines needs to be involved (e.g., hydrology, engineering, land use planning, policy studies, sociology, anthropology, economy, law, and others) as well as expertise on the respective local and regional situation through the inclusion of local knowledge and understanding possible development ("futures") by collaborative

modelling with stakeholders. This is particularly the case when it comes to the development of adaptation strategies and measures and their implementation.

*Spatial scales*

The genesis of a flood event is dependent on various spatial factors on different scales, such as shape of the catchment, precipitation pattern and structure of land use and land cover (Bronstert, 2003; Kundzewicz et al., 2014). Furthermore, the interaction of adjacent

sub-catchments and upstream and downstream dynamics determine whether the incident becomes a disaster. In contrast to the natural spatial boundary conditions in the formation of floods, the management of flood risk is mostly organised at the administrative level (Evers and Nyberg, 2013). The way in which the different administrative levels interact depends on the administrative and governance structure. These different spatial scales relevant to flood risk research require a multi-layered and interlinked system of spatial levels and boundaries. This includes administrative borders, catchment borders, local affectedness, regional,

national, and transnational policy levels and organizations.

*Temporal scales*

Different temporal scales and dynamics play an important role for more reliable flood risk research. Specifically for DRR, the different temporal scales relate to the different phases before, during, and after the event. Di Baldassarre et al. (2018) explain that particular attention needs to be paid to assessments, which support long-term decisions need to be considered. They state further

that „the useful inclusion of adaptation in the assessment of future risks requires—among other things—knowledge about the change in (property-level) adaptation over time" (Di Baldassarre et al., 2018, p. 15). Scenario techniques which simulate future conditions are helpful in FRM. Thieken et al. (2016) distinguished a reference scenario that reflects the present situation, from a baseline scenario, which shows future (transient) behavior without adaptation, and alternative scenarios, which quantify the effects of different adaptation or management options. Analogous to this, scenarios and projections of possible future dynamics of flood

generation processes, including their uncertainties, should be considered (Kundzewicz et al., 2018). Accordingly, risk assessments





need a clear concept of the temporal dynamics or pathways which should be explored. Risk assessment of flood events include a consideration of hydrology, climate, weather, mid- and long-term climate change aspects, history and development of legal frameworks, societal and economic settings, and situations, the evolution and persistence of cultural and historical aspects and other aspects should be considered. Rapid development and the uncertainty with respect to future development requires that future

studies should employ methods which address uncertainties of knowledge and values, e.g. participatory scenario development, back-casting, collaborative development of strategies (Höllermann et al., 2025; Kok and van Vliet, 2011; Quist and Vergragt, 2006; Heijden, 2005). Thus, it is essential to consider various and diverging temporal patterns, paces, and processes.

### *Situatedness*

Flood risk and vulnerability are highly context-specific and situated in the specific historical, cultural, and social setting of the

respective situation. This is especially relevant for North-South-collaboration as this often relates to either multiple geographies, or knowledge is produced through the collaboration of a multi-cultural team. Thus, the situatedness of the respective flood risk research needs to be taken into account for co-creation of knowledge. Ziga-Abortta and Kruse (2023) identified a broad set of factors explaining the institutional context of FRM in Ghana, including socio-cultural, socio-political, legislative-regulatory, and fiscal-economic drivers that can be used to characterise the specific situation of vulnerability to flood in Ghana. De Brito et al.

(2018) found in a case study in Brazil that the perception and assessment of vulnerability varies greatly depending on the perspective of the stakeholders involved and their experiences and professional background. For example, the assessment of criteria such as gender, income, or technical flood protection varied depending on the experience and perspective of the stakeholders and thus influenced the vulnerability assessment accordingly. As a consequence, vulnerability criteria have to be adjusted and specified, resulting in a different flood risk characterisation related to the situatedness of the observed case. The consequence for TD research

projects in FRM is a thorough stakeholder analysis, which is crucial for addressing and including diverse stakeholders in the co-production of knowledge processes. Joint identification and prioritisation of criteria that address, e.g., social vulnerability, adaptive capacity or institutional settings are important since these are situated in a specific social, historical, political and spatial context (Juhola and Kruse, 2015).

### *Technical tools and methods*

The complexity of floods, including spatio-temporal scale challenges and the importance of situatedness, calls for socio-technical tools and methods for visualisation and communication of flooding with stakeholders to, e.g., address the assessment of flood risk or adaptation capacity (Jonoski and Evers, 2013). In scenario development there is a general need for advanced technical methods and tools for data collection (remote, on-site, real-time, historical timelines, etc.) and processing of data for hydrological and hydro-dynamic modelling which include aspects of Land Use Land Cover Change (LULCC) or climate change effects. Furthermore,

socio-economic data, qualitative ethnographic data or data derived from empirical social science methods needs to be integrated in socio-hydrological frameworks, models, and scenarios. However, we suggest the application of a TD and co-production approach in generating results from the latter methods and sources, as this goes a step further, hereby connecting to co-designing and social learning processes. This is supported by Jafarzadegan et al. (2023) who suggest to enhance our understanding of flood-generating mechanisms and recommend inter alia a coupling of hydroclimatic and hydrological perspectives. Furthermore, they

recommend transitioning to an innovative earth system modelling approach that connects meteorological, hydrologic, hydrody-namic, and decision-making components and allows for feedback exchange. We think that, in addition to this, human influence should also be integrated into the coupling approach since the human factors have a reciprocal influence on the flood risk system (Evers et al., 2017). Kundzewicz et al. (2018) promote considering scenarios and projections of possible future dynamics of flood generation processes, including their uncertainties. Evers et al. (2016) and Almoradie et al. (2023) found – based on case studies

in London, Hamburg, and Togo/Benin – that collaborative modelling is supportive in social learning processes. Modelling different



flood risk-related situations and scenarios is often used as a tool, yet it is so complex that scientific experts from other disciplines as well as practitioners, decision-makers, and citizens often view it as a black box (Almoradie et al., 2023; Evers et al., 2016). Geo data, measured/in situ hydrological data, remote sensing, socio-economic data, and land use data play an important role but also increase the complexity and potential uncertainty of the research results. Thus, trust in data and methodologies is core when it comes to TD methods in flood risk research, validation, and groundtruthing of data and models as well as improving the data base. This requires a transparent presentation and explanation of the data, the processes and the models used as well as of the capabilities and limitations of the models and socio-technical tools. Otherwise, the usage of the socio-technical tool will not be accepted and thus not be helpful for the process (Jonoski and Evers, 2013; Evers, 2008).

Table 1 illustrates the characteristics of flood risk, the need for flood risk research and their implications for transdisciplinary co-production of knowledge.

**Table 1: Characteristics of flood risk and implications for transdisciplinary flood risk research**

| Important aspects in Flood risk research | Characteristics of flood risk research | Implications for the design of transdisciplinary co-production of knowledge |
|---|---|---|
| Complexity | Various interrelated factors influence flood risk within and across environmental and societal systems, leading to complexity of flood risk problems | Broad and cross-cutting interdisciplinary and practical expertise<br>Collaboration of scientist and non-scientists |
| Spatial scales | Different spatial scales are mutually dependent and influence flood risk | Consideration of different spatial scales and administrative levels and inclusion of representatives from different spatial levels and institutions |
| Temporal scales | Different temporal dimensions result in different levels of flood risk and require targeted action | Short-, mid- and long-term aspects should be considered<br>Application of participatory scenario development to understand feedback loops, mechanisms, and potential entry points for flood risk reduction<br>with stakeholders who represent local / expert knowledge for identifying potential interventions |
| Situatedness | Flood risk and vulnerability are highly context-specific and situated in the specific historical, cultural, political, social and special setting of the situation | Knowledge from different stakeholders as well as views and values and assessments should be incorporated in all phases of the FRM research process<br>Joint identification of criteria for vulnerability and risk assessment and their respective relevance should be agreed upon |
| Technical tools | Hydrological and hydrodynamic flood modelling are key methods of investigation for understanding and depicting processes and interrelationships as well as possible development options<br><br>A lack of data or level of detail influences the informative value and increases uncertainties, especially at a local level | Include (advanced) socio-technical tools which can help to understand the complex system, visualise potential management options, data and functionalities, and effects in scenario development<br><br>Limitations of data and tools should be transparent<br><br>How these are to be used should be agreed upon with the stakeholders |



**2.2 Characteristics of North-South collaboration and consequences for the design of transdisciplinary research projects**

TD projects that are realised in North-South collaboration differ from research projects that are conducted in the Global North (Pärli et al., 2022a; for an overview on specific aspects of North-South research cf. table 2). Scholars relate this to a different

research tradition in North-South collaboration where, already in the 1970s, *inclusive and participatory research methodologies* were adopted and replaced traditional, top-down approaches to knowledge production. Through the inclusion of different types of knowledge, (e.g. of local stakeholders) and the participation of diverse actor groups incorporating equal partnership, the research projects proved more effective in both the co-production of a robust and sound knowledge basis as well as in the uptake of the research and actual impact creation (Hirsch Hadorn et al., 2006; Pärli et al., 2022a).

Further, North-South research projects have a strong focus on the practical *applicability of results* as they often have a problem-oriented approach and aim at developing sound solutions (Pärli et al., 2022a). The comparative analysis of Pärli et al. (2022a) shows that North-South projects report more on societal effects that are actually relevant and applicable for the stakeholders and the context-specific uptake of knowledge, while the effects of North projects were more related to knowledge production and tangible research outputs such as academic publications.

Due to the additional transnational and intercultural dimension, North-South research projects offer opportunities for *mutual learning*. Through the appreciation of diverse forms of knowledge, they advance mutual understanding among North-South partners and, through this, capacity building and exchange for all involved partners (Pärli et al., 2022a, Hirsch-Hadorn et al., 2006).

An additional important aspect of North-South research projects is that they often come along with *power imbalances* that can complicate collaboration and partnership. The power imbalances can, for example, be related to funding regulations where funding

and project coordination are managed by organisations in the global North or other implicit hierarchies that can negatively impact the effective and efficient implementation of projects. Developing power-sensitive designs of TD collaboration as well as installing mechanisms for balancing power are thus of specific relevance for the TD co-production of knowledge in North-South projects (Pärli et al., 2022a; Schneider et al. 2023).

Last but not least, *differences in research cultures* among different partners from North and South call for translational efforts to

facilitate TD co-production of knowledge in North-South partnerships. Accordingly, creating situations for training on sensitivity to cultural differences and reflective capacities can enhance mutual understanding and collaboration (Pärli et al., 2022a).

**Table 2: Characteristics of North-South collaboration and their implications for transdisciplinary co-production of knowledge (own compilation following Pärli et al. 2022, Schneider et al. 2023, Hirsch Hadorn et al., 2006;)**

| Important aspects of North-South research | Characteristics of North-South collaboration | Implications for the design of transdisciplinary co-production of knowledge |
|---|---|---|
| Inclusive/participatory approaches | Awareness of the need for the inclusion and participation of local actors | Participatory methodologies and equal partnership in research collaboration |
| Applicability of results | Tend to have a strong focus on the practical applicability, societal effects, and the uptake of knowledge of results | Orientation towards real-world problems and the development of sound and sustaining solutions |
| Mutual learning | Appreciation of diverse forms of knowledge to advance understanding among the different North and South partners | Create opportunities for joint learning, capacity building, and exchange for all partners |



| Power imbalances | Power imbalances, funding structures, and implicit hierarchies can complicate collaboration implementation | Efforts from the start to reduce power imbalances; develop power-sensitive designs of transdisciplinary collaboration; install balancing mechanisms |
|---|---|---|
| Cultural differences | Translation issues and differences in research cultures can complicate the research process | Train mutual understanding and sensitivity to cultural differences |

### 2.3 Mechanisms for impact generation by transdisciplinary research

The characteristics of both flood-risk related research and North-South collaboration have implications for the design, implementation, and effects of TD tools and methods for the co-creation of knowledge. At the core, these characteristics call for generating impact by research for the respective field of study. On a meta level, Schneider et al. (2019) studied 31 case studies from TD projects in order to identify their theories of change through common patterns and lessons learnt and to define important mechanisms to generate impact. The analysis also included some projects from the Global South. Among the 31 projects were 10 case

studies from Asia, 10 from Africa, six from Latin America, and nine global. From this, Schneider et al. (2019) identified three generic mechanisms for impact generation through the TD co-production of knowledge: a) promoting systems, target, and transformation knowledge for more informed and equitable decision-making, b) fostering social learning for collective action, and c) enhancing competencies for reflective leadership. They found that the three generic mechanisms were activated by seven recurrent strategies, such as access to information or advice and training. The three mechanisms can be seen as fundamental processes in

which, through the involvement of different actor groups, new knowledge, shared understanding, and new competencies are co-produced and materialised in concrete activities or outcomes. These three mechanisms can create impacts in the form of more informed and equitable decision-making, collective action, and reflective leadership. Schneider et al. (2019) pointed out that their research provides evidence that the most promising pathways to impact are long-term, adaptive processes that combine elements of the three mechanisms of impact generation in parallel or over time. They conclude that the question is not which mechanisms

or strategies are better than others but in what situation and combination they might be most effective.

In this context, we reflect on our approach and the experiences we gained within the TD Flood Risk Research project in Ghana, the PARADeS project, against the background of the identified generic mechanisms with interrelated components for TD co-production of knowledge.

In the following chapter we describe the PARADeS project, its methodological approach, and the methods applied. With this, we

juxtapose the three generic mechanisms with the findings from the PARADeS project.

### 3 Transdisciplinary co-production of knowledge in Flood Risk Research - implementation and experiences from the PARADeS project Ghana

#### 3.1 The PARADeS project

Ghana is one of the countries in West Africa that is most prone to floods (Aggrey, 2015; Amoateng et al., 2018; World Bank,

2011) with devastating effects, especially for the urban poor (Okyere et al., 2013). In recent years, many parts of Ghana have experienced extreme floods affecting about one million people (Adegoke et al., 2019; IFRC, 2017). The Floodlist database lists 33 extreme flood events in Ghana during the last 20 years. Besides the annual occurrence of major floods, Ghana also experiences cascading disasters triggered by floods, which disrupt critical infrastructure.



Studies on the future of floods in Africa show a likely increase in extreme flood risk in Western Africa (Hounkpè et al., 2022). It
has been estimated that about 12% of the African population face food insecurity after 2020, with various cascading effects such
as increasing levels of poverty, loss of lives, and conflicts (IPCC, 2021; Gil, 2022). Flood Disaster Risk Management (FDRM)
regimes and sustainable adaptation approaches are thus urgently needed.

To tackle this, the Ghanaian government has established several policies to reduce flood impacts, such as the National Water Policy
(Government of Ghana, 2007) or the Blue Agenda (Addo and Danso, 2017), which address flooding and related threats by focusing
on public education and the enforcement of building regulations. However, there is still a knowledge gap with respect to detailed
hazard and flood dynamics and means of sustainable implementation of measures to reduce flood risk (Almoradie et al., 2020).

As part of the PARADeS project, which was funded by the German Ministry for Education and Research, we developed and tested
a TD multi-method approach for the co-production of knowledge on flood risk disaster dynamics and mechanisms as a core basis
to identify and implement proactive protection and behavioural measures and increase competence for sustainable FRM. The aim
was to contribute towards enhancing Ghana's national flood disaster risk reduction and management strategy by investigating key
drivers and pressures, existing flood risk and disaster management, governance and policy, and human–water interaction, and by
developing scenarios, action plans, and feasible and sustainable flood risk prevention measures.

A participatory mixed-method approach comprising hydrological and hydrodynamic modelling, participatory mapping, question-
naires, workshops, focus group discussion, system dynamic modelling, and the analysis of vulnerability, including failure of critical
infrastructure, was employed for three case study areas in Ghana. The dynamics of human–flood-interaction were identified to-
gether with practitioners, and adaptation measures were identified in a participatory manner.

Ghanaian project partners were the Water Resources Commission (WRC), the National Disaster Management Organization
(NADMO), and the West African Science Service Center on Climate Change and Adapted Land Use (WASCAL). The German
consortium consisted of the University of Bonn, University of Freiburg, University of Applied Sciences Magdeburg and the Flood
Competence Centre (for further information please see the PARADeS website (https://www.geographie.uni-bonn.de/parades/de).
During the project, there was intensive cooperation and continuous exchange amongst all partners.

Before the actual start of the research project, there was a six-month preliminary study to conceptualise the project and write the
project proposal. Besides various interactions with partner institutions and stakeholders, core elements of the project were three
workshop series in Ghana in the different case study areas. The first was conducted in a hybrid format due to Covid restrictions.
The other two workshop series took place in Ghana. With this approach, we aimed to make our research actionable, and to design
and implement knowledge translation mechanisms. The project content and work stages are described in more detail in Evers et
al. (2023b).

### 3.2 What worked? Exemplifying a transdisciplinary co-production process in the PARADeS project

In the following, we describe and reflect on the TD co-production of flood risk-related knowledge in the PARADeS project. We
do this along the three key mechanisms for impact generation as identified by Schneider et al. (2019): knowledge production,
social learning, and competence building (see section 2.3.). However, we would like to add to these mechanisms three mediators:
first, the joint formulations of research questions, second, building trust and creating ownership, and third, the anchoring of project
results. For Walter et al. (2007) motivation, joint understanding, increased trust, and network building are such mediators, which
they also regard as a relational outcome of co-production processes. Even though we agree that mutual trust and understanding are
relational outcomes of such an approach, we argue that it is also important to deliberately initiate ground for those mediators. Since
the nature of TD research includes joint framing of problems and goals for constitutive transformative change (Lang et al., 2012),
we started the co-production processes with the joint formulation of relevant research questions and research design. Furthermore,



we paid extra attention to and invested in building trust and creating ownership throughout the entire project. Another mediator that requires co-evolution with ownership is the anchoring of results. From our perspective, these three mediators were crucial for

our TD approach, as they helped ensure that the key mechanisms of co-production / TD research are functioning. Thus, we enhanced the conceptualisation of the generic mechanisms of impact generation by highlighting the three mediators. Figure 1 illustrates the interplay of those mediators and the generic key mechanisms as applied in the PARADeS project.

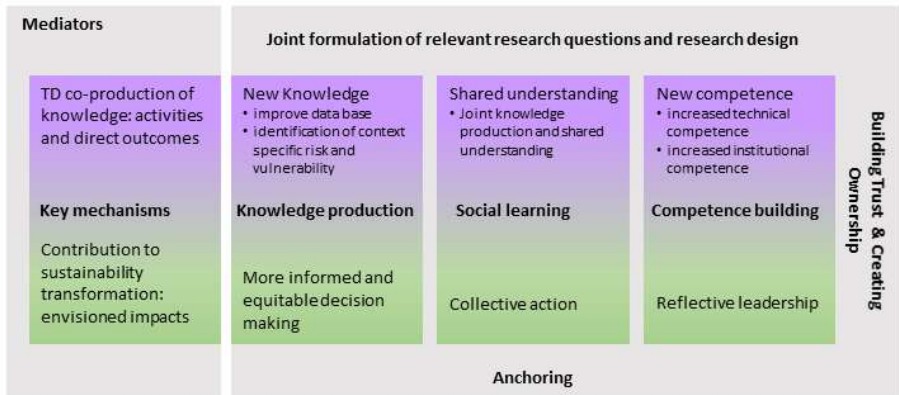

**Figure 1: Overview of the generic key mechanisms of impact generation through TD research (knowledge production, social learning**
**and competence building) with mediators as central elements through the entire course of the project (trust & ownership), at the beginning of TD activities (joint research questions), and as a prerequisite for impact generation (anchoring).**

In the following sections we elucidate in more depth the concrete collaborative steps, elements and methods of our PARADeS project as a best practice example. We start with the three key mechanisms (section 3.2.1), followed by the central mediators (section 3.2.2) we identified as essential along our research process. Both mechanisms and mediators supported the co-production

of knowledge throughout our project activities and direct outcomes, and enabled the envisioned transformative impacts of the project in the long run (cf. figure 1).

### 3.2.1 Key mechanisms for impact generation

**Knowledge production**[1]

*Improving the data base*

Data collection is a core basis for a FR research project. This includes hydrological, land use, socio-economic data as well as data about risk perception, awareness and governance processes. In the co-production of knowledge, TD methods are combined with modelling and fieldwork methods in order to improve data bases. For example, in order to generate flood hazards and flood risk maps for the case study areas, input data such as rainfall data, discharge data, cross-sectional data, and land use information were collected from data bases and the respective institutions in Ghana to set up hydrological, hydrodynamic and risk cascade models.

Additionally, Ghanaian partners did field work to collect cross-sectional data. Furthermore, we conducted a web-based participatory mapping process during the first workshop series for all case study areas in order to obtain information on flood hot spots from the local and regional experts and thus to improve and validate the model results. In our case, the model results showed a

---

[1] While Schneider et al. (2019) use knowledge promotion as a term, we prefer knowledge production since we think that the creation of new knowledge is central for the co-production process.



high level of agreement with the regional expert knowledge as well as with observed data during recent flood events, thus proving the suitability of our models inter alia through the expert judgement of the stakeholders. In addition to improving the database

through the co-production of knowledge, this procedure also built trust and improved the acceptance of the model results.

In data-scarce regions, citizen science is another way of improving the data base through the co-production of knowledge. In our case, we developed a pilot project on the collection of water level and discharge data with the help of community gauges (Meyer et al. 2024). The pilot has shown high potential not only for data availability but also for awareness raising in the respective communities. However, we also learnt that certain aspects such as timing or coordination of different sub-activities are crucial and

should be taken into account to make citizen science projects a success.

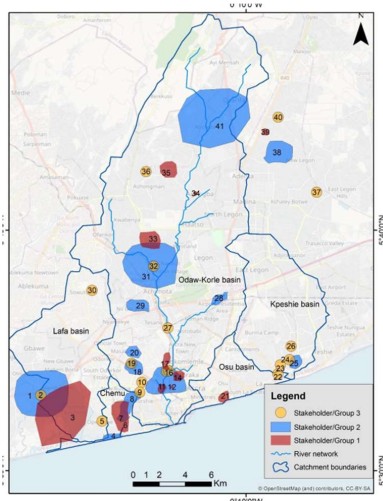

**Figure 2: Results from collaborative modelling of flood hot spots © PARADeS project**

With respect to the cascading impact of the models on critical infrastructure, expert interviews and specific workshops with experts proved to improve the models significantly.


### *Identification of context-specific risk and vulnerability*

As indicated in section 2.1 vulnerability and risk related to flood are highly context specific, and thus perception also varies (de Brito et al., 2018). Therefore, together with our project partners and the people potentially affected by floods we generated joint knowledge on the perception of vulnerability and risk; we hence complemented the information on flood hazard and risk derived

from the hydrological and hydrodynamic models. In workshops we engaged and exchanged with different groups of stakeholders, experts and decision makers such as residents, local government representatives, national disaster managers and hydro-meteorological experts. With the help of online voting tools (e.g. mentimeter) we asked them to choose their preferred adaptation measures; this proved to be highly dependent on their individual flood perception, experience and the degree to which they have been affected. Further, flood causes, impacts, and adaptation measures were identified using the problem-tree approach. This method helped to

jointly carve out the underlying causes of flooding problems in the three case study areas and to show how they relate to land use, urban and rural development, behavioural issues, inadequate drainage systems, and waste management problems. Exchange with locally affected communities was arranged in the form of focus group discussions with representatives of communities and organised by the Ghanaian project partners (esp. NADMO). Another methodological tool for co-producing knowledge in the specific risk and vulnerability context was the analysis of stakeholders and their interaction with each other in the form of social networks.



This helped to understand how stakeholders differ in perspectives, interests, capacities and resources as well as knowledge and ideas about FRM and how they interact and exchange with each other. The social networks were depicted based on qualitative interviews and validated in additional stakeholder workshops, where the analysis was presented, critically discussed and adjusted based on the stakeholders' feedback (Ziga-Abortta 2025).

**Social learning**

*Joint knowledge production and shared understanding*

Social learning starts with the organising partners. Thus, joint field visits and mutual visits of the participating research and non-academic partners were an important way of creating mutual understanding at eye-level. This involved visits and return-visits of stakeholders and researchers in Ghana and Germany with the aim to share both pilot case examples as well as challenges and difficult situations when managing flood risk. Beyond show casing, also experiencing daily working environments of the respective

institutions increased mutual understanding and reflection of differences and similarities in culture, social-political, and institutional settings.

Joint knowledge production, however, extends beyond the participating project partners and includes a diverse set of affected stakeholders and community representatives. Participatory scenario development is often recognised as a valuable approach for addressing socio-ecological problems that are characterised by intricate, unpredictable, and uncertain challenges such as flood risks

(Kok and van Vliet, 2011). It not only integrates different perspectives and mobilises a broad variety of knowledge but also creates shared understanding among the stakeholders, thus helping to facilitate credibility and mutual learning for planning and future response (Evers et al., 2016). In the following, we describe our scenario development as one example of how we initiated knowledge production and shared understanding.

In PARADeS, the participatory scenario development followed a participatory back-casting approach (Quist and Vergragt, 2006)

and focused on the most important factors for the future of FDRM in Ghana. We followed a three-step approach: i) identification of causes, impacts and adaptation measures, ii) developing shared understanding of problems and desirable futures in FDRM, iii) developing scenarios and robust pathways.

i) In a first hybrid workshop series with stakeholders in 2021, causes, impacts, and adaptation measures were identified using the

problem-tree approach. Based on the problem tree, participants were able to test which variable (causes and impacts) was targeted by the adaptation measures and how these measures translated through the system. In the first workshop, it became clear that a combination of hard and soft measures is needed to effectively reduce flood risk; however, many of the suggested measures focused on short-term perspectives. During the second workshop, when applying a scenario exercise the measures tended to be more transformative and long-term.

ii) These workshops, helped in generating a common understanding of problems, desirable futures, policy options and adaptation measures in FDRM using a 30 year time frame. Regarding actionable outcomes, the participating stakeholders were asked to develop policy options they considered necessary and plausible. They thus produced a policy toolbox addressing different policy levels from local to national and supranational, and considering different policy instruments from regulatory and financial to informational, educational, and collaborative instruments (Kruse et al., 2023; Evers et al., 2023b; Kok and van Vliet, 2011).

iii) In the third step, building on the shared problems, the participants developed a set of different scenarios and robust pathways. The scenarios were built using an impact/uncertainty grid to convey critical uncertainties. Critical uncertainties are those whose impact is assessed as very high and whose uncertainty is also high. Two of those uncertainties were commonly chosen to create a 2x2 scenario matrix where both uncertainties cover a range from positive to negative development, thereby creating four explorative scenarios. In moderated group work following the world-café method the participants subsequently discussed the policy and



adaptation options of each scenario which could lead to the identified desirable future. This activity resulted in robust pathways towards improved FRDM taking into account local context. In this second round of workshops the continuous exchange among the participants showed an effect on the assessment of potential measures, which became apparent in a distinct change of preferred adaptation measures (Höllermann et al. 2025).

Thus, our recommendation is to engage stakeholders throughout the whole co-production process and regularly, through work-
shops, to ensure that the various activities can translate into learning. This requires providing space for the discussion of previous results with the stakeholders, as described above. Additionally, there are distinct differences between urban and rural and between stakeholder and affected population priorities and necessities for adaptation measures, which shows that sensitivity to situatedness is also important in the organisation of the co-production process (cf. sec. 2.1 on situatedness).

Beyond this learning within activities, the insights from the workshop interactions, for example the identification of adaptation
measures (e.g. adapted land use planning, wetland protection, and the creation of retention ponds and dams) were also key for the improvement of the hydrological and hydro-dynamic models. This helped in the development of scenarios which are regarded as plausible and was thus useful in ensuring that the jointly developed Flood Information System (FIS) gained acceptance.

**Competence building**

*Increase technical competence*

In order to be able to apply the co-produced knowledge and experience in a targeted manner, we implemented a number of activi-
ties, such as training sessions as part of workshops, production of online materials for self-learning and more. In the following we describe a few examples of the many activities.

We designed and developed the Flood Information System (FIS) for the Competence Building according to the wishes and re-
quirements of the end users. We carried out various training sessions with the web-based flood information system. A PARADeS
project member was on site in Accra at the partner institution for several days to transfer the FIS to the planned Flood Early Warning Centre. In addition to this, materials were designed and developed to provide further training. These materials were made available online. They were developed in accordance with the Open Education Resource (OER) standard formats so that they are generally available and usable. For capacity building, we developed and published online courses explicitly for students as teachers in order to transfer expertise, knowledge and skills and also to use this group as potential multipliers.

Together with our Ghanaian partners we published a handbook to make the results of the project available not only in scientific publications, but also for decision-makers and experts from the field. For practitioners we prepared an online and freely accessible publication on the topic of lessons learnt.

*Increase institutional capacities*

To increase institutional competences, a set of different TD methods can be applied in order to create outcome and impact in co-
production processes and support reflective leadership for effective FRM. A second method that was developed and applied in the PARADeS project was training to strengthen collaborative governance between FRM stakeholders. In a workshop format, both existing individual as well as collective network activities were depicted and analysed with the result that each participant learned about their own collaboration and options for improving collaboration within their network. An open online course in OER format
was subsequently developed to also enable training of the networking technique independently within the different organisations (Ziga-Abortta and Kruse, 2023).

Last but not least, the development of policy briefs in TD teams which took up specific results and translated them into differenti-
ated and applicable recommendations proved to be relevant for the envisioned outcome and impact. Here, the integration of expe-
rience and knowledge of both research partners as well as experts and decision makers in Ghana and Germany proved important



to identify the specific topic for which policy briefs were developed as well as to find the adequate language and communication channels for bringing the recommendations to the specific target groups.

A summary of the above-described core mechanisms is juxtaposed in table 3. Table 3 also highlights the corresponding methods applied to conduct the respective step as well as an illustration of the envisioned outcomes and impacts of the PARADeS approach.

**Table 3. Mechanisms and sub-elements for the co-production of knowledge in Flood Risk research identified by Schneider et al. (2019)**
**and applied/modified in the PARADeS project**

| Mechanisms for co-production adapted from Schneider et al. 2019 | Mechanisms and sub-elements according to flood risk research applied in PARADeS project | Transdisciplinary methods in PARADeS project (exemplified) | Envisioned outcome and impact for PARADeS project |
|---|---|---|---|
| **Knowledge promotion** | **Knowledge co-production** | | |
| New knowledge* | Improve data base | ▪ Surveys<br>▪ Community gauges (citizen science)<br>▪ Participatory mapping<br>▪ Focus group discussions<br>▪ Field work / mapping by partners | ▪ Validation & groundtruthing of data and models which increases the quality and increases trust in data and models, contributing to more and useful information |
| | Identification of context specific risk and vulnerability | ▪ Hazard modelling<br>▪ Hydrological and hydrodynamic and risk cascade models<br>▪ Participatory mapping<br>▪ Focus group discussions<br>▪ Survey<br>▪ Network analysis<br>▪ Validation workshop for institutional vulnerability | ▪ Validation & groundtruthing of data and models which increases the quality and increases trust in data and models<br>▪ The co-production leads to more equitable and informed decision making |
| **Social learning** | **Social learning** | | |
| Shared understanding | Joint knowledge production and shared understanding | ▪ Participatory scenario analysis<br>▪ Exercise with Flood Information System (FIS) on scenarios and effects of interventions<br>▪ Visit and return-visit of stakeholder and researcher in the respective Organizations in Ghana/Germany | ▪ Understanding complexity of Human–flood interaction and root causes<br>▪ Identifying interventions for collective action<br>▪ Creating mutual understanding through an outside perspective and shift in perspectives respectively supporting collective action |
| **Competence building** | **Competence building** | | |
| New competence | Increase technical competence | ▪ Elicitation requirements for the FIS in a workshop<br>▪ Training on web-based FIS, which includes a flood early warning system<br>▪ Production of training material and trainings of specialist, decision and policy makers and students | ▪ Tailored FIS to needs of users<br>▪ Competence building on the usage of the FIS<br>▪ Capacity development on technical knowledge to increase the potential for usage |
| | Increase institutional capacities | ▪ Policy toolbox<br>▪ Training for collaborative governance<br>▪ Training on collaborative governance and network building<br>▪ Raise awareness regarding potential institutional challenges | ▪ Delineate variety of policy choice to decision makers<br>▪ Enable networking outside established collaborations increasing reflexive leadership<br>▪ Target-group oriented communication supports uptake in decision making processes<br>▪ Policy briefs |



\* Schneider et al. 2019 regard "new knowledge" as the outcome of the TD process. In our perspective, knowledge production is part of the TD process and the new knowledge is integrated into feedback loops with social learning and competencies.

### 3.2.2 Importance of mediators for impact generation

Mediators play a crucial role in enhancing the generic key mechanism of impact generation in TD research. They facilitate the processes of knowledge production, social learning and competence building, ensuring that transformative impacts are effectively embedded within and beyond the project. In our approach, we highlight three key mediators (cf. figure 1): first, the joint formulation of research questions and research design at the beginning of the TD activities, second, building trust and creating ownership throughout the project, and third, the anchoring of project outcomes as a prerequisite for generating impact.

**Joint formulation of relevant research questions and research design**

It was important for us to include a collaborative phase in which the research project was conceptualised and the application written prior to our actual project work. During this phase, the research questions were developed together with stakeholders, and the case study areas were jointly defined. Joint research questions and goals represents a mediator which strongly co-evolves with trust building. Together with our practical partner institutions we jointly identified the research questions for the research project and discussed and co-designed the research proposal. The actual initiative for the research proposal was started by the African partners.

The basis for concrete formulation of the research topics was thorough literature research and a data survey. We had the opportunity to conduct this during a six-month pre-study which was financed by the funding agency. During this phase, we were able to conduct, as a core step, a first stakeholder analysis. Based on this we held a workshop with stakeholders and conducted focus group discussions in order to identify the main research issues and to conduct a continuous exchange with our three main project partners to write the research proposal. To obtain the first insights into the current FRM situation, a web-based questionnaire was prepared

and conducted in the different case study areas in the country. Using the problem tree method the participants of the workshop then discussed the gaps in FRM in Ghana as identified in the online survey by (Almoradie et al., 2020). In addition to the joint formulation of research questions, we found it of utmost importance to identify the case study areas together with the stakeholders. Based on focus group discussions and participatory mapping exercises we identified three case study areas for the main project phase looking at, on the one hand,neuralgic areas with regular flooding with high damage and on the other hand at different flood

types (fluvial, pluvial and coastal flooding) that includes Accra, Kumasi and Volta region/Bolgatanga (Fig. 3).

Our experience shows that this step of a joint formulation of research questions and research design creates a crucial basis for TD research and ownership even before the actual start of the project and thus increases the chance of successful co-production processes and finally the implementation of the envisioned impacts.



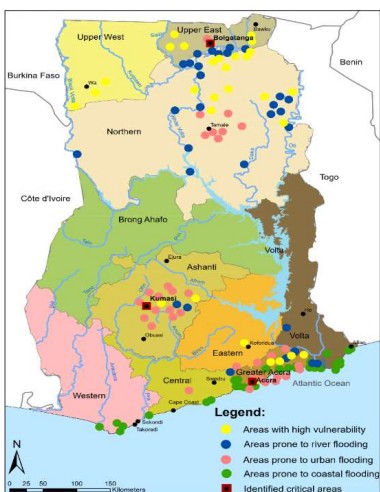

**Figure 3: Participatory mapping of case study areas © de Brito (2018)**

**Building trust and creating ownership**

Although mutual trust and understanding is a relational outcome of a TD approach, we paid particular attention to and invested in
building trust and creating ownership throughout the project. Throughout the entire duration of the project, our experience was that
openness to situational and cultural/social necessities or 'standard procedure' was supportive. Being aware that the results of the

project would only be realised if they met with interest and acceptance, it was essential that there was cooperation based on trust,
which is not a one-time check box, but a process. Therefore, we very consciously implemented and observed the following aspects.
In addition to the stakeholder workshops with their co-production of knowledge and mutual learning, PARADeS initiated an ex-
change of high- and mid-level experts and administrative staff from disaster management organisations in Germany and Ghana.
The same applies to the academic exchange between universities, which included students, PhD students, as well as mid- and high-

level academics. Both exchanges helped to intensify collaboration between the respective organisations as well as to foster under-
standing of the specific perspectives, expertise and situational setting of FDRM in Germany and Ghana. More generally, through-
out the project we ensured a reliable presence and responsiveness, even at higher hierarchical levels.

All activities described in the previous sections (sub-section of 3.2.1) were conceptualised and carried out in order to create own-
ership. A few examples are described in more detail in the following. A central element of creating ownership was that the FIS

was tailored to the needs of the end users. The FIS, developed for the catchments Odaw (Accra), Aboabo (Kumasi) and the White
Volta, is a web-based application to support and improve decision-making capabilities for FRM. It brings together scientific data
produced by the models and local knowledge obtained through inputs from stakeholders. Stakeholder input was obtained from two
stakeholder workshops where they were involved in the validation of hazard results and the requirements, needs and design of the
FIS. We elicited the needs and wishes during one of these workshops with participants from a broad range of different institutions,

created a draft for the architecture design and requirements of the tool which was again agreed among the stakeholders. Only then,
after receiving feedback from the stakeholders, did we construct the FIS.

The joint formulation of policy briefs and a documentation of lessons learnt from the project, the handbook and OER materials,
were instigated at the express request of the practice partners, thus adding valuable and tangible outcomes while simultaneously
creating ownership. To further strenghten ownership events were organised at which the presidents of the respective partner or-

ganisations could publicly affirm their commitment to the project, supported by media coverage.



**Anchoring project results**

Further, in order to create outcomes that can be implemented beyond the scope and timeframe of the project, PARADeS aimed early in the development of the project to prepare the transferability and continuation of the results. For example, with the completion of the design and implementation of the FIS under the PARADeS Project, a Memorandum of Understanding (MoU) was

signed between the University of Bonn and the partner institution, the Water Resources Commission (WRC), where the FIS is implemented and will also be used in the future. It is also planned that the FIS will be integrated into the Flood Early Warning System (FEWS) Centre hosted by WRC. The University of Bonn team supported the setting up and updating of the FIS, and provided capacity building and technical advice to FEWS-Accra. This also helped to create both ownership and to develop a structure for the sustainable implementation of the FIS within the newly established FEWC.

Another anchor was established by developing the Ghana Flood Label, a tool which specifically addresses homeowners and aims to identify potential flood-risk reduction measures for their property. The development of this tool was carried out in co-operation with NADMO, the official authority in Ghana for civil protection. Also, a training unit on the the Ghana Flood Label was held with NADMO employees. Additionally, we collaborated with the University of Cape Coast, which, although not an official project partner, was willing to work with the PARADeS project to develop a use case to test the applicability of the flood label. The

application was implemented and evaluated by university students as part of a study project. These measures not only led to quality assurance and ensured applicability but also to the training over the course of the project (via NADMO and the University of Cape Coast) of multipliers for future application of the Flood Label. In the policy briefs and the handbook, which represented a central joint output of the PARADeS project, we formulated consequences for policy and planning, not only in the case study for Ghana in the Global South but also for the Global North or the funding policy of the donor.

We believe that these efforts are key for TD research providing support for the anchoring of project results beyond the project and in contributing to triggering transformative change.

Thus, in addition to the key mechanism of co-production, we identified the joint formulation of relevant research questions, the creation of trust and ownership, and the embedding of project outcomes as central processes throughout the TD research journey. We see these elements as mediators within a complex process that not only facilitate collaboration, but also enable the generation

and implementation of results by addressing the complexity of the issue at hand.

## 4 Discussion and conclusion

### 4.1 The project design: the characteristics of transdisciplinary flood risk research in North-South collaboration

PARADeS was conceived as a TD project that was initiated jointly by practice and scientific partners. The project had several scientific objectives: Besides identification of flood risk and flood risk measures on different temporal and spatial scales for a data-

scarce region and modelling risk cascades for critical infrastructure we aimed to understand which drivers influence institutional vulnerability and how strategies and measures of FRM are implemented effectively with regard to collaborative governance. By producing knowledge in these fields, we aimed at impact generation beyond the time scope of the project. A further objective was the development of a generic TD approach that can be replicated or adapted elsewhere based on our research design for TD FR research in Ghana.

Building on other TD projects, literature and the project partners' experience in Europe and West Africa, we jointly developed the presented approach, structure and concept. Importantly, the method design was realised in close coordination with diverse stakeholders in an interdisciplinary research team. As illustrated in Chapter 2 and table 1, the characteristics of flood research were considered and their implications for the TD research design were identified.





Within our flood risk research in PARADeS, a key aspect of the TD co-production of knowledge was the consideration of different temporal and spatial scales. This included employing hydrological and hydraulic models, carrying out thorough stakeholder analysis, and conducting workshops and focus group discussions that made it possible to integrate not only a broad range of expertise but also diverse stakeholder perspectives throughout the research process. Among other methods, these goals were achieved through participatory scenario development, collaborative identification of vulnerability criteria and focus group discussions with community members.

In order to ensure that the co-produced knowledge is also readily available beyond the project, we developed a web-based flood information system and offered associated training courses for all user groups. The visualisation of interrelationships and the consequences of interventions in the existing flood systems in three case study areas supported enhanced understanding of the complex systems and illustrated the effect of different management options.

In our North-South collaboration, we prioritised participatory methodologies and equal partnerships as these are crucial for inclusive approaches (see table 2). Obeying these criteria, we also addressed concrete real-world problems ensuring the applicability of results. We fostered joint learning and capacity development. To tackle power imbalances, we implemented mechanisms, aiming at minimising them (e.g., joint visits), and we remained mindful of cultural differences by actively raising awareness and incorporating these nuances into our work, e.g. by regular feedback sessions.

It became clear that the mechanisms as identified by Schneider et al. (2019) for generating impact also play a central role in our project. The elements of co-production of knowledge, social learning and competence building are reflected in the structure and research design of the PARADeS project. For example, we supplemented the co-production of the knowledge with specific activities and direct results that are particularly relevant to flood research. This includes, for example, the improved database and identification of context-specific risk and vulnerability indicators, as well as the concretisation of what needs to be specified in joint knowledge production. In terms of new competence, we differentiate between increased technical competence and increased institutional capability. Both are needed to ensure uptake and implementation of new knowledge. Accordingly, we have illustrated the methods we used to achieve this in table 3.

We exemplified in more detail the respective methods and tools that were used in order to achieve the envisioned outcomes. We believe that the general research design and utilisation of the various methods can be used as an example in other research projects and work as a blue print. In addition to these key mechanisms, we identified three mediators that we see as central to promoting the co-production of knowledge and to ensuring the success for the implementation of research results. Going forward, it would be valuable to reflect on the combination of these key mechanisms and the role these mediators play in further research projects, drawing lessons learnt from that reflection (see section 4.2).

## 4.2 Key mechanisms and mediators: the conceptual design for transdisciplinary co-production of knowledge

In their review, Schneider et al. (2019) identified co-production of knowledge, social learning, and competence building as the three key mechanisms of impact generation through TD research that enable a shift from co-producing knowledge to impact generation through activities and direct outcomes for sustainable transformation. These mechanisms also played a central role in the PARADeS project. They are instrumental in structuring the project and especially in helping generate not only scientific results but also outcomes that are directly relevant for the practitioners and hold the potential for creating long-term impact.

Even though the key mechanisms of impact generation are foundational for TD research, from our point of view they require additional enablers to ensure implementation and sustainability of project outcomes. We conceptualise these enablers as mediators that facilitate and reinforce the processes of knowledge production, social learning, and competence building, ensuring that transformative impacts are effectively embedded within and beyond a project's lifespan. In our approach, we highlighted three key





mediators: (1) joint formulation of research questions and research design at the outset of transdisciplinary activities, (2) building trust and creating ownership throughout the project, and (3) anchoring project outcomes as a prerequisite for generating lasting

impact.

The combination of key mechanisms and mediators can provide valuable lessons for future TD projects. Figure 1 shows how the corresponding mechanisms and mediators interact and complement each other. For example, understanding the co-evolution of mechanisms and mediators across scales, sectors, researchers and stakeholder groups, can maximise stakeholder engagement and strengthen outcomes. This process can foster broader systemic change and contribute to sustainable transformation.

An important dimension of this TD approach is its context-sensitive nature, a lesson we could learn from North-South cooperation (see Table 2). For example, certain institutional or cultural contexts may require longer periods of trust building or more targeted methods of participatory problem framing, especially in places where there is scepticism regarding external influence. In other contexts, approaches such as integrating traditional knowledge or partnering with community leaders may prove essential to create ownership and subsequently provide a basis for anchoring results. In this respect, our multi-level approach (from community

leaders to national policymakers) proved supportive in managing potential spatial and institutional misfits.

In addition, as we have shown with our workshop series, iterative learning plays a crucial role in allowing for reflection of new knowledge while building trust in each other and in each other's expertise. This shows how the key mechanisms work together with the mediators. These insights help in choosing methods of interaction and in refining the extent to which stakeholder input is sought, e.g., during problem framing or when responding to emergent group dynamics, with the aim of further strengthening the

effect of the mediators. Without building trust, knowledge co-production is at risk. By documenting such decisions and their effects, future TD projects can identify which strategies best promote the synergy between co-production, social learning and competence building.

Many projects fail to achieve lasting impact or progress. Even when stakeholders acquire new technical and institutional skills, if these skills are not embedded in a supportive policy or organisational framework, project momentum can fade. For example, we

used co-development of action plans and policy briefs as strategies for embedding project outcomes in relevant institutions. Our training on collaborative governance further helped to ensure that newly acquired capacities are integrated into long-term practice. More generally, the analysis of successes and challenges in embedding can provide recommendations for scaling up or replicating similar TD efforts in other regions or sectors.

The interplay between the three mechanisms of impact generation and the three identified mediators represents a promising TD

approach for enhancing the effectiveness, relevance and sustainability of TD research outcomes.

### 4.3 Products, outcomes, impacts: Learning from TD research in North-South collaboration

A literature review on TD co-production of knowledge in North and North-South projects comes to the conclusion that North projects score higher on products, that is, tangible outputs such as academic publications or outreach material (Pärli et al., 2022a). The latter also report, similar to other authors, that there is a reported trade-off: the production of knowledge and the involvement

of stakeholders are often seen in conflict with each other (also Jahn et al., 2012; Chambers et al., 2021; Newig et al., 2019). This cannot be backed by our experiences and reflections on the transdisciplinary co-production of knowledge in the PARADeS project. There was a strong focus on inclusion and participation as well as on producing tangible and practical outcomes for the Ghanaian stakeholders and case study regions. Yet, much knowledge was produced that is of relevance beyond the actual case in Ghana, particularly with respect to the methodologies and models. A great deal of academic and outreach material was produced , both

targeting academia in Ghana and internationally as well as practitioners, stakeholders, and flood disaster experts.



One important element means of attaining both scientific and practical outputs as well as the envisioned long-term impacts was the early and continuous stakeholder identification and inclusion. This was achieved both by scientific application of Social Network Analysis (SNA) as well as practical and self-reflective network analysis (cf. section 3.1) from the start to the very end of the project. This helped to understand how stakeholders differ in perspectives, interests, capacities and resources as well as knowledge

and ideas about FDRM and, as an important side effect, create trust and ownership and enable meaningful anchoring for (potential) long-term impacts. The importance of creating stakeholder involvement processes that accounts for diverse and often diverging perspectives throughout the entire project is evident.

In this way, this aspect can set the basis for collaborative governance, where FRM is not only developed and implemented through government agencies, but also actively includes stakeholders from non-governmental and civil society organisations, private com-

panies, and the local population. Collaborative governance is regarded as a key approach to improve Flood Disaster Risk planning and management as it helps to coordinate meaningful community involvement that helps to both improve FDRM and create risk awareness and self-efficacy; it facilitates the building of a robust framework of accountability and transparency that helps to build public trust; and it fosters inclusivity and collaboration across various stakeholders, also aiming to improve just resource distribution within FDRM processes (Ziga-Aborrta and Kruse 2023).

The involvement of stakeholders in collaboratively identifying scenarios and measures, along with community group discussions through social learning, has offered a platform to grasp diverse perspectives, interests, and local knowledge. It helped determine the heterogeneity of needs between rural and urban areas but also within the urban areas. This not only facilitated the research project's activities but also enriched their comprehension and awareness of the issues, causes, effects, and concerns. This approach helps to initiate participation of local communities in supporting regionally adapted FDRM.

Last but not least, what can we learn from the experiences of this North-South project PARADeS for North projects? In Europe, for example, we can see that very interesting research is being carried out and is also being put into practice in the form of publications or the publication of flood risk maps. In the case of an emergency, many examples of past disasters and flood catastrophes in Europe and worldwide have shown that there is a lot of knowledge in theory, but the actual flood risk is not minimised or not minimised to the extent possible on the basis of the theoretical background. We can also gain a lot for research projects in the

North from the characteristics of North-South collaborations, such as the inclusion and participation of local stakeholders, the focus on the practical applicability of results or creating opportunities for mutual learning..

### 4.4 Conclusion

In FRM, there is a large gap between theoretical knowledge and the actual implementation of research findings. The consequence of this is that we know in theory how we can reduce risk. Yet, this knowledge is not consistently implemented. In severe extreme

events with high damage and casualties, we often hear, that those responsible could not have known the extent of this flood, or that the effectiveness of planned measures was perhaps not clear or the implementation of measures was not feasible due to a lack of acceptance.

Accordingly, similar to other fields of sustainability transformation, research on FRM and DRR needs to consider the challenges of producing a sound knowledge base, and through this supporting the development of robust decisions and the implementation of

strategies and action plans in FRM. TD research, specifically the co-production of knowledge, is often considered to be a means of helping to overcome or at least to tackle these challenges.

With the PARADeS project, we have illustrated an example of how such a TD project in FR research can be carried out using the tools and methods appropriate for the specific settings of a North-South collaboration. We learnt from this empirical case that the mechanisms of knowledge production, social learning and competence building identified by Schneider et al. (2019) are just as



relevant to the project as the application of three mediators: (1) joint formulation of relevant research questions and the co-design of the research project, (2) trust building and creating ownership before and during the whole project, and (3) anchoring of project results to increase the potential of implementation and foster contribution to sustainability transformation. Applying these in the conceptual research design of flood risk research enables us to meet the three challenges in DRR: (1) the need for a sound knowledge base on which to build robust decisions, (2) developing strategies and measures that address both existing risks and

consider inherent uncertainties of knowledge and (3) the implementation of effective FRM strategies and action plans. Indeed, our research provides evidence that a promising pathway to creating the envisioned impact in flood risk research is to combine the three mechanisms of impact generation and the mediators in parallel and over time. Hence, we conclude that the question is not which mechanisms or strategies are better than others, but in what situation and combination they might be most effective. Thus, we emphasise the need for projects to further elucidate the interplay of the key mechanisms and mediators in a context-sensitive

manner so as to improve our ability to produce tailor-made approaches.

*Author contributions.* ME and SK conceptualized the research and wrote the original draft of the paper. ME, BH and SK wrote the manuscript. All authors conceptualised the developed framework (figure 1), BH and ME designed mainly figure 1. ME finalised the first draft of the manuscript.

*Competing interests.* The contact author has declared that none of the authors has any competing interests.

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
