# Peer review of "Transdisciplinary co-production of knowledge for effective flood risk management"

_EGUsphere, 2025_

## Referee Comment (RC1)

Dear authors,

I appreciate the work you describe in the manuscript, as I consider TDR very important and it is crucial to reflect on these processes. I find it extremely useful that you align flood risk research components with practical TDR advice. Moreover, the article made me very curious about the project; it seems you have had a privileged opportunity to conduct TDR research. However, I do have some concerns that I believe need to be resolved prior to publication.

First, reporting and reflecting on a transdisciplinary research project in Ghana without involving Ghanaian partners in these reflections (co-authorship) seems inappropriate given the pluralism and inclusiveness prerogative of knowledge co-production.

Second, I miss a clear research gap identified in the introduction that highlights the concrete contribution of the paper vis-à-vis the literature you describe. You write that you expand upon a research area aiming to address three objectives, but those, in comparison with the brief state of the art, are not indicative of the relevance of your contribution.

Third, the rationale of the state of the art remains unclear to me vis-à-vis the reflections you promise. You spend a lot of space on the characteristics of flood research and North–South collaboration, but barely introduce your key component of reflections later—the impact mediators.

Similarly, you barely focus on the framework you introduce (Figure 1), even though you propose this to be an innovation that could be used in similar work. (This leads back to the lack of introducing a research gap and reviewing other TDR frameworks.)

Fourth, in your reflections you describe the methods you use in the project vis-à-vis ensuring the impact. Given that your aim is to reflect on TDR and knowledge co-production, I would like to see much more detail about how you actually managed the equal partnership, the trust-building, and ongoing reflections. I am missing this throughout. E.g., you mention that you organized exchanges between countries and people at different levels, and joint field visits that created mutual understanding at eye level, or you mention a TD team at some point, and community discussions elsewhere. All of this sounds great but does not tell us how you made sure that this actually created trust and a feeling of equality, and how it enhanced learning. Reading about a TDR project, continuous reflection is crucial, but there is little to no information about this in the manuscript at the moment.

- I really liked where you mention that experiencing each other's work environment increased mutual understanding; this goes in the direction I am speaking of.

All of this information is crucial to better understand power imbalances and cultural differences you might have encountered and addressed—two of the characteristics for TDR you mentioned in Table Two that are probably the most distinguishing factors for North–South collaboration.

**Within the state-of-the-art sections**, there are a couple of things that I think could be improved, apart from dedicating more space to TDR and impact mechanisms:
First, you spend much space on the characteristics of flood risk research and only at the end provide a table that links it to TDR. This link is crucial and should be highlighted and built into the text. Indeed, the

links seem to be drawn out of nowhere in the table (lacking references) and a better introduction of what TDR is. In a way, you do this in the next section on North–South collaboration, but not systematically.

Second, the section on North–South collaboration to me reads a bit strange—in a way as if TDR research is fundamentally different in North–South collaboration. While I agree that there are differences such as those presented by Pärli, some formulations in your text I stumble across. For example, only in the first sentence do you say TD projects, later you always speak of research projects, and I think it would help to stick with TDR, as this is what the research you cited focused on as well. Also, it would help if you stuck with comparative language: e.g., writing "North–South projects have a strong focus on the practical applicability of results…" implies that other research projects do not, which is not true. Similarly, when you write "North–South research projects offer opportunities for mutual learning." Sure, this is true—but also for other TDR projects. Finally, I think Table Two highlights a list of characteristics for all TDR projects, where some aspects will be more critical for North–South collaboration. I would appreciate it if you changed the tone a bit.

**Inconsistencies and lack of clarity here and there:**

- **e.g., Table Three:** You have a column of TD methods where you list hazard modelling and surveys among others, which are certainly not TD methods—unless you redesign them or embed them in a process and facilitation that enables co-production. This is also the case in places in the text, where only in some instances you show how you created a TD setup for the implementation of the method. Actually, in the cell next to it you write the co-production leads to more equitable and informed decision-making—which is a very bold claim that I do not think you can substantiate with the paper as is.

- **e.g., in paragraph 355:** You all of a sudden write "we thus recommend…". First, I find that strange to have a single recommendation at this point and one that is not particularly new at all. Also, I do not agree that just engaging throughout will lead to learning—could you be more specific here? And finally, you mention here that this requires space for the discussion of previous results with stakeholders, as described above (but I could not find any reference to that).

Overall, I had a hard time reading the article, and I have to admit, given time constraints, I was not able to carefully read the discussion and conclusions section, which is also why my comments do not include that section. I wrote the review by myself, but will, before I finish, use AI to fix typos and check for any illegible sentences, which sometimes happen when I write fast under time pressure. I hope this will help you read my comments.

I really believe that you did some excellent work, but I think for TDR and knowledge co-production reflections the content needs to be shifted a bit—maybe with some more illustrative examples—to actually show what it takes to make FR research transdisciplinary. I know it's not an easy feat, when the article is already quite long.

---

## Author Comment (AC1)

**RC2 Anonymous Referee #2**

**First** of all, I would like to ask you can you provide a definition of how you define co-production; knowledge; social learning; and institutional capacities (difference to social capacity).

*Thank you for your valid comment. We will add a subsection in the Introduction defining key concepts and will elaborate more on the concept in chapter 2.*

**Secondly**, what's actually your theoretical framework you used for your study? At the moment, it's a little bit unclear about the main used theoretical concept.

*Many thanks for this comment. We think we can present Chapter 2.3 more clearly as a conceptual framework, expand it and perhaps separate it from 2.1 and 2.2.*

**Another question** is: what's actually new – or what we don't know so far from TD projects across the globe; especially section 3.2.

*Thank you for your comment. We will strengthen the introduction in the revised version to highlight the research gap. Specifically, we will point out that while frameworks for TD research exist (e.g., Lang et al., 2012; Schneider et al., 2019), their application in flood risk management within North–South collaborations has rarely been systematically examined. Our paper contributes by applying and expanding Schneider et al.'s impact-generation framework. Specifically, we identified three mediators (joint formulation of research questions, trust and ownership, and anchoring of results) that are crucial in North–South TD settings. These two points represent a concrete conceptual and methodological contribution.*

2. is something we already know and discuss for many years in different TD projects across the globe. How you deal with the challenge of North-South collaboration; especially in the sense of the ongoing decolonial discourse in different disciplines?

*Many thanks for this comment. When revising the introduction and section 2, we will add the discourse on co-production of knowledge (e.g., Castelli et al. 2025* https://www.tandfonline.com/doi/full/10.1080/02626667.2025.2571065#d1e1550*) and the plurality of knowledge referring also to the debate on de-colonizing   referring to Chilisa 2017, French et al. 2024, Shackelton et al. 2023. and Zonta 2023.* [i]

An important question of TD projects lies on the question about the impact: what's the actual impact of the project and second question lies on the long-term perspective of the TD process within the region; are there any hints for a long-term/institutionalised living lab in the region? Or does the TD process end with the project?

*We will expand Section 3.2.2 (Anchoring results) and Section 4.3 (Products, outcomes, impacts) and will work out the specific results more clearly.*

Another question reflects how are non-scientific actors involved within the overall research process, such as framing problem, analyzing problem, exploring impact; how you integrate both realm (science and practice) within you project?

*Thank you. yes, we can work out the specific results more clearly.*

How did you organise and manage the reflection process within the project and can you extend this part within your paper (results section)?

*Reflection was institutionalised through iterative workshops, stakeholder feedback sessions, and joint evaluation of intermediate outputs after the second workshop. We can work out the reflection process more clearly in the manuscript.*

Finally, I would like to ask you what's actually new of your paper in terms of theoretical discourse as well as methodologically in sense of TD research.

*The theoretical novelty lies in extending Schneider et al.'s framework with mediators, providing a refined model of TD impact generation in North–South collaborations. We think that these examples can serve as a starting point for reflecting on the implementation of mechanism and mediators in future TD Flood Risk Research. We are happy to work out this more clearly in the manuscript.*

[i] Chilisa, B. (2017). Decolonising transdisciplinary research approaches: An African perspective for enhancing knowledge integration in sustainability science. Sustainability Science, 12(5), 813–827. https://doi.org/10.1007/s11625-017-0461-1

French, M. A., Barker, S. F., Henry, R., Turagabeci, A., Ansariadi, A., Tela, A., Ramirez-Lovering, D., Awaluddin, F., Latief, I., Vakarewa, I., Taruc, R. R., Wong, T., Davis, B., Brown, R., & Leder, K. (2024). Responsible north–south research and innovation: A framework for transdisciplinary research leadership and management. Research Policy, 53(7), 105048. https://doi.org/10.1016/j.respol.2024.105048

Shackleton, S., Taylor, A., Gammage, L., Gillson, L., Sitas, N., Methner, N., Barmand, S., Thorn, J., McClure, A., Cobban, L., Jarre, A., & Odume, O. N. (2023). Fostering transdisciplinary research for equitable and sustainable development pathways across Africa: What changes are needed? Ecosystems and People, 19(1), 2164798. https://doi.org/10.1080/26395916.2022.2164798

Zonta, A. L., Jacobi, J., Mukhovi, S. M., Birachi, E., Groote, P. V., & Abad, C. R. (2023). The role of transdisciplinarity in building a decolonial bridge between science, policy, and practice. GAIA - Ecological Perspectives for Science and Society, 32(1), 107–114. https://doi.org/10.14512/gaia.32.1.7

---

## Author Comment (AC2)

**RC1**: Susanne Hanger-Kopp**

Dear authors,

I appreciate the work you describe in the manuscript, as I consider TDR very important and it is crucial to reflect on these processes. I find it extremely useful that you align flood risk research components with practical TDR advice. Moreover, the article made me very curious about the project; it seems you have had a privileged opportunity to conduct TDR research. However, I do have some concerns that I believe need to be resolved prior to publication.

**First**, reporting and reflecting on a transdisciplinary research project in Ghana without involving Ghanaian partners in these reflections (co-authorship) seems inappropriate given the pluralism and inclusiveness prerogative of knowledge co-production.

Thank you for this important comment. Of course, throughoutt the whole course of the projects we had regular reflection with all partners and including both academic as well as non-academic partners from North and South which was essential for the research design and project results. Indeed, for the development of this manuscript we did not include our African partners because of limited time recourses on their part. Furthermore, they were more interested in other formats of publications, e.g. policy briefs and lessons learned for non-academic audiences. A list of joint publications are found at the end of this document.

However, after reflecting your very valuable comment that it appears quite inappropriate to reflect on a transdisciplinary North-South collaboration in a small and purely academic co-authorship, we have come to the conclusion that for the revision we will invite two practice partners, who initiated the project by approaching us and contributed most to the project (Charlotte Norman from NADMO and Dr. Mawuli Lumor from WRC, Ghana), as well as the two researchers, who particularly promoted and facilitated the transdisciplinary North-South collaboration (Dr. Adrian Almoradie, EGLV and Dr. Joshua Ntajal, University of Bonn, Germany), as co-authors. They will be engaged actively in the revision process. Many thanks for this important point of reflection.

**Second**, I miss a clear research gap identified in the introduction that highlights the concrete contribution of the paper vis-à-vis the literature you describe. You write that you expand upon a research area aiming to address three objectives, but those, in comparison with the brief state of the art, are not indicative of the relevance of your contribution.

Thank you for your comment. We will strengthen the introduction in the revised version to highlight the research gap. Specifically, we will point out that while frameworks for TD research exist (e.g., Lang et al., 2012; Schneider et al., 2019), their application in flood risk management within North—South collaborations has rarely been systematically examined. Our paper contributes by applying and expanding Schneider et al.'s impact-generation framework. Specifically, we identified <a href="three-mediators">three-mediators</a> (joint formulation of research questions, trust and ownership, and anchoring of results) that are

crucial in North–South TD settings. These two points represent a concrete conceptual and methodological contribution.

**Third**, the rationale of the state of the art remains unclear to me vis-à-vis the reflections you promise. You spend a lot of space on the characteristics of flood research and North–South collaboration, but barely introduce your key component of reflections later—the impact mediators. Similarly, you barely focus on the framework you introduce (Figure 1), even though you propose this to be an innovation that could be used in similar work. (This leads back to the lack of introducing a research gap and reviewing other TDR frameworks.)

Many thanks for this observation. Based on your comment, we noticed that we have not introduced and described our conceptual framework (based on Scheider et al. and others) and its components based on the state of the art sufficiently. In the revised version of the manuscript, we will revise Section 2.3 and introduce the key components of our conceptual framework as well as Figure 1 to better link the state of the art to our reflections.

**Fourth**, in your reflections you describe the methods you use in the project vis-à-vis ensuring the impact. Given that your aim is to reflect on TDR and knowledge co-production, I would like to see much more detail about how you actually managed the equal partnership, the trust-building, and ongoing reflections. I am missing this throughout. E.g., you mention that you organized exchanges between countries and people at different levels, and joint field visits that created mutual understanding at eye level, or you mention a TD team at some point, and community discussions elsewhere. All of this sounds great but does not tell us how you made sure that this actually created trust and a feeling of equality, and how it enhanced learning. Reading about a TDR project, continuous reflection is crucial, but there is little to no information about this in the manuscript at the moment.

Thanks a lot for this valid comment! We will provide in the revised version concrete examples to describe further how equal partnership, trust-building, and ongoing reflections were fostered.

Within the state-of-the-art sections, there are a couple of things that I think could be improved, apart from dedicating more space to TDR and impact mechanisms: First, you spend much space on the characteristics of flood risk research and only at the end provide a table that links it to TDR. This link is crucial and should be highlighted and built into the text.

Thank you for your comment. Yes, we will elaborate more on how the characteristics of flood risk research and TDR are linked.

Second, the section on North–South collaboration to me reads a bit strange—in a way as if TDR research is fundamentally different in North–South collaboration. While I agree that there are differences such as those presented by Pärli, some formulations in your text I stumble across. For example, only in the first sentence do you say TD projects, later you always speak of research projects, and I think it would help to stick with TDR, as this is what the research you cited focused on as well. Also, it would help if you stuck with comparative language: e.g., writing "North–South

projects have a strong focus on the practical applicability of results..." implies that other research projects do not, which is not true. Similarly, when you write "North–South research projects offer opportunities for mutual learning." Sure, this is true—but also for other TDR projects. Finally, I think Table Two highlights a list of characteristics for all TDR projects, where some aspects will be more critical for North–South collaboration. I would appreciate it if you changed the tone a bit. Inconsistencies and lack of clarity here and there.

Thank a lot, the comment is very helpful and gives us the opportunity to show once again from a different perspective how we have implemented the methods and approaches that are important to the FRR in a TDR process, thereby also addressing your comment above.

1 We have published a couple of other papers and documents together with them such as:

- Almoradie, A., Brito, M. de, Evers, M., Bossa, A., Lumor, M., Norman, C., Yacouba, Y., J. Hounkpe (2020): Current flood risk management practices in Ghana: Gaps and opportunities for improving resilience. Journal of Flood Risk Management doi: 10.1111/jfr3.12664 ...

- Evers, M., Almoradie, A., Ntajal, J., Höllermann, B., Johann, G., Meyer, H., Schüttrumpf, A., Kruse, S., Ziga-Abortta, F., Bachmann, D., Schotten, R., Lumor, M., Norman, C., & Adjei, K. (2023). Pro-active flood risk managment using a transdisciplinary multi-method-approach (Nos. EGU23-11980). EGU23. Copernicus Meetings. https://doi.org/10.5194/egusphere-egu23-11980

- Evers, M., Almoradie, A., Ntajal, J., Höllermann, B., Johann, G., Meyer, H., Kruse, S., Ziga-Abortta, F., Bachmann, D., Schotten, R., Lumor, M., Norman, C. & Adjei, K. (2024). Lessons Learned from PARADeS Project for Flood Disaster Risk Planning and Management in Ghana. https://doi.org/10.48565/bonndoc-195

- Lumor, M., Norman, C., Almoradie, A, Evers, M (2024): Flood Early Warning System and Centre, In: Evers, M., Kruse, S., Almoradie, A. & M. Tuschen (Eds.) (2023): Managing Flood Disaster Risk in Ghana. Findings, Products and Recommendations. https://doi.org/10.6094/UNIFR/242726

- Evers, M., Delos Santos Almoradie, A., Ntajal, J., Höllermann, B., Johann, G., Meyer, H., Kruse, S., Ziga-Abortta, F., Bachmann, D., Schotten, R., Lumor, M., Norman, C., & Adjei, K. (2024). Lessons Learned from PARADes Project for Flood Disaster Risk Planning and Management in Ghana [Report]. Department of Geography, University of Bonn. https://doi.org/10.48565/bonndoc-195